# Boron Clusters as Enhancers of RNase H Activity in the Smart Strategy of Gene Silencing by Antisense Oligonucleotides

**DOI:** 10.3390/ijms232012190

**Published:** 2022-10-13

**Authors:** Damian Kaniowski, Katarzyna Kulik, Justyna Suwara, Katarzyna Ebenryter-Olbińska, Barbara Nawrot

**Affiliations:** Centre of Molecular and Macromolecular Studies, Polish Academy of Sciences, Sienkiewicza 112, 90-363 Lodz, Poland

**Keywords:** boron cluster, RNase H, antisense oligonucleotide, ASO, EGFR, BNCT, B-ASO

## Abstract

Boron cluster-conjugated antisense oligonucleotides (B-ASOs) have already been developed as therapeutic agents with “two faces”, namely as potential antisense inhibitors of gene expression and as boron carriers for boron neutron capture therapy (BNCT). The previously observed high antisense activity of some B-ASOs targeting the epidermal growth factor receptor (EGFR) could not be rationally assigned to the positioning of the boron cluster unit: 1,2-dicarba-*closo*-dodecaborane (0), [(3,3′-Iron-1,2,1′,2′-dicarbollide) (1-), FESAN], and dodecaborate (2-) in the ASO chain and its structure or charge. For further understanding of this observation, we performed systematic studies on the efficiency of RNase H against a series of B-ASOs models. The results of kinetic analysis showed that pyrimidine-enriched B-ASO oligomers activated RNase H more efficiently than non-modified ASO. The presence of a single FESAN unit at a specific position of the B-ASO increased the kinetics of enzymatic hydrolysis of complementary RNA more than 30-fold compared with unmodified duplex ASO/RNA. Moreover, the rate of RNA hydrolysis enhanced with the increase in the negative charge of the boron cluster in the B-ASO chain. In conclusion, a “smart” strategy using ASOs conjugated with boron clusters is a milestone for the development of more efficient antisense therapeutic nucleic acids as inhibitors of gene expression.

## 1. Introduction

Antisense oligonucleotides (ASOs) have been widely developed for the treatment of various human diseases, including those that were previously incurable. Following the approval of the first ASO drug, Fomivirsen, in 1998, the U.S. Food and Drug Administration has approved ten ASO-based drugs to date. The first antisense drug was designed to inhibit the translation of mRNA encoding key proteins of the early region of cytomegalovirus and was used for the treatment of retinitis caused by this patogen [1]. In 2004, the FDA approved Macugen (Pegaptanib), an RNA aptamer targeting the vascular endothelial growth factor (VEGF) isoform primarily responsible for the pathologic ocular neovascularization in age-related macular degeneration (AMD) and for vascular permeability [2]. In 2013, the FDA approved the use of Kynamro, also known as Mipomersen, as an agent that targets mRNA of apolipoprotein B100, for the treatment of familial hypercholesterolemia (FH) [3]. Eteplirsen was later introduced for the treatment of Duchenne muscular dystrophy (DMD) and Nusinersen was approved in 2016 for the treatment of spinal muscular atrophy [4,5]. Eteplirsen binds to the DMD-relevant exon 51 of dystrophin RNA and allows exon 52 to splice to exon 51, resulting in a truncated but partially functional protein (exon switching strategy) [4]. Nusinersen also applies an exon-switching strategy to increase the amount of full-length functional survival motor neuron 2 protein (SMN2). After hybridization with its target, this oligonucleotide forces the incorporation of exon 7 into mRNA and prevents the formation of short-lived/nonfunctional proteins [5,6]. On 30 March 2016, the FDA approved Defibrotide (now known as Defitelio, Jazz Pharmaceuticals), which is the drug intended for the treatment of severe hepatic veno-occlusive disease (sVOD) that occurs after high-dose chemotherapy and autologous bone marrow transplantation [7]. Later, in October 2018, the FDA and Health Canada also approved Tegsedi (Inotersen), a chemically modified antisense oligonucleotide that inhibits hepatic production of transthyretin (TTR) in adults [8]. Golodirsen was approved in 2019 and Viltepso (Viltolarsen) in 2020 for DMD patients carrying mutations amenable to exon 53 skipping [9,10]. The following year, 2021, the fourth ASO drug, amondys 45 (Casimersen), was approved for the treatment of DMD. It is used to treat patients who have genetic mutations amenable to skipping exon 45 of the Duchenne gene [11].

More recently, ASO have been used to combat severe acute respiratory syndrome coronavirus 2 (SARS-CoV-2), which is the cause of the current 2019 coronavirus pandemic (COVID-19) causing a global health emergency. The most advanced ASO approach is to knock down angiotensin-converting enzyme 2 (ACE2) which is the entry receptor of SARS-CoV-2. Using splice-switching oligonucleotides (SSOs), it should be possible to modulate alternative splicing of the ACE2 gene to remove critical regions required for SARS-CoV-2 entry into host cells [12,13,14].

FDA-approved ASOs based on the RNase H mechanism include such drugs as Mipomersen, Inotersen and Volanesorsen, the latest approved by the European Medicines Agency (EMA) in 2019 for the treatment of familial chylomicronemia syndrome (FCS) [15].

ASOs are short synthetic single-stranded oligo-2′-deoxyribonucleotides (13–25 nucleotides), that have the potential to modulate the expression of target genes commonly associated with the development of a variety of diseases [16]. Several mechanisms by which small oligonucleotides can be used to regulate gene expression in human cells are known. ASOs function through Watson–Crick hybridization by directly binding to complementary sequences of messenger RNAs (mRNAs) and modulating their function [12,17]. Although most oligonucleotide therapeutics function on gene silencing via mRNA-targeted ASOs [12] similar to short interfering RNAs (siRNA) [18] and microRNAs (miRNA) [19], other strategies are being pursued, including alternative splicing modifications to include or exclude mRNA exons [20,21], or increasing translation through targeted interaction with upstream open reading frames (uORFs) [22].

The most commonly used antisense mechanism involves recruitment of the ribonuclease H (RNase H) to the heteroduplex consisting of the single-stranded ASO (DNA) and the mRNA of the target protein, followed by cleavage of the RNA within such a duplex. This endoribonuclease is able to recognize any DNA/RNA heteroduplex longer than 5 base pairs [23]. Once this structural motif recognition has occurred, RNase H hydrolyzes the internucleotide bond of the RNA, resulting in two truncated RNA products (containing a 5′-phosphate and 3′-hydroxyl termini) and releasing the intact and active ASO oligonucleotide strand [6].

Human cells express two types of RNase H: RNase H1 and RNase H2. Human RNase H1 is active as a single peptide, while RNase H2 is a heterotrimeric enzyme [17,24,25]. Scientific evidence shows that despite a lower expression level only RNase H1 is involved in the ASO mechanism in mammalian cells. This is probably because RNase H2 is tightly bound to chromatin, while RNase H1 is found in the nucleus, cytoplasm and mitochondria [23]. The two enzymes also differ in terms of their cofactor requirements and activity. The RNase H type 1 has been shown to be activated by both Mg^2+^ and Mn^2+^ or in the presence of sulfhydryl reagents, whereas RNase H type 2 is activated only by Mg^2+^ and inhibited by Mn^2+^ or sulfhydryl reagents [25]. Furthermore, the use of oligonucleotides enables precision and/or personalized medicine approaches, as ASOs can theoretically be designed to selectively target any gene with minimal, or at least predictable, side effects [26]. In addition, there are known methods to improve the design of ASOs that address the problem of off-target safety, sequence- and chemical-dependent toxicity, degradation by nucleases, or targeted drug delivery [27].

New “smart” chemical strategies are constantly being sought to enhance the activity and efficacy of therapeutic nucleic acids. Recently, boron clusters have received special attention because of their beneficial biomedical functions as pharmacophores [28,29], “enhancers” of drug action [30,31,32,33], membrane carriers [34,35,36,37], protein binders [38,39] and receptor ligands [31,32,40,41].

We reported that single-stranded or nanoparticle-embedded antisense oligonucleotides decorated with boron clusters (B-ASOs) inhibit epidermal growth factor receptor (EGFR) [42,43,44,45] and boron cluster-modified short interfering RNAs (B-siRNAs) inhibit beta-secretase (BACE) protein biosynthesis [46]. In addition, our study showed that B-ASOs decorated with metallacarborane (FESAN) clusters increased the rate of snake venom phosphodiesterase (svPDE)-catalyzed DNA oligonucleotide hydrolysis [38]. Furthermore, we have shown that the free FESAN molecule has a high affinity for crude venom protein mixtures as well as for single-stranded DNA and acts as an interface (the “glue”) between the two reacting biomolecules [38].

Due to their chemical structure, boron clusters resemble 3D capsules rich in boron atoms, which are required for boron neutron capture therapy (BNCT) [47,48]. In BNCT, ^10^B boron atoms are introduced into cancer cells. Upon irradiation with a thermal neutron beam, they capture neutrons and become ^11^B isotope. The ^11^B boron atoms are highly unstable and split into the highly linear energy transfer species (LET) ^4^He and ^7^Li, which thermally destroy the cancer cells in which they are generated. In addition, B-ASO may have a dual effect: on the one hand, they provide a therapeutic platform for BNCT, and on the other hand, they reduce the mRNA of EGFR by antisense activity, thus decreasing the radio-resistance of cancer cells [49,50,51].

In the present study, we focused on the effects of different types of boron clusters conjugated with antisense oligonucleotides, their structure, charge, and position in the ASO chain on the activity of the endoribonuclease H. These studies provide a basis for a better understanding of the role of boron clusters conjugated to ASO on RNase H activity in antisense RNA cleavage. The proposed chemical modification of ASO drugs offers innovative advantages in the development of B-ASO drugs that would act more effectively in the cellular and in vivo systems.

## 2. Results

### 2.1. Chemical Synthesis of Boron Cluster-Conjugated Oligonucleotide Models

Two 13-nt reference oligonucleotides DNA **1** and **2** (a homopyrimidine and a purine rich oligomers, respectively, Table 1) were designed to be *iso*-sequential with two fragments of a previously explored 22-nt antisense oligonucleotide (ASO-22) targeting EGFR mRNA [38,42,43]. Five congeners of **1** (**P1a**, **P1b**, **P1c**, **P1d**, **P1e**), three congeners of **2** (**P2a**, **P2b**, **P2c**, Table 1) carrying a 2′-*O*-propargyluridine (U_Pr_) moiety and their unmodified complementary 13-nt oligonucleotides RNA **3** and **4**, respectively, were obtained by automated solid-phase synthesis.

Alkyne-functionalized oligonucleotides **P1a**–**P1e** and **P2a**–**P1c** were post-synthetically modified with boron clusters [(3,3′-Iron-1,2,1′,2′-dicarbollide)^-1^] (N_3_-alkyl-FESAN), [Fe(C_2_B_9_H_11_)_2_]^-^ (**B1**), 1,2-dicarba-*closo*-dodecaborane (N_3_-alkyl-1,2-DCDDB, [C_2_B_10_H_12_]) (**B2**), or dodecaborate (N_3_-alkyl-DDB), [B_12_H_12_]^2-^ (**B3**), which were endowed with an alkyl azide (N_3-_alkyl) functional group suitable for copper-catalyzed azide-alkyne cycloaddition (“click chemistry” reaction mode), (Figure 1).

The resulting oligonucleotides **1a_1_**, **1a_2_**, **1a_3_**, **1b_1_**, **1c_1_**, **1d_1_**, **1e_1_**, and **2a_1_**, **2b_1_**, **2c_1_** containing a uridine moiety labeled with FESAN, 1,2-DCDDB, or DDB (**U_B1_**, **U_B2_** or **U_B3_**, respectively, in Table 1) were isolated by RP-HPLC and characterized by MALDI-TOF MS or electrospray ionization mass spectrometry (ESI MS). Oligonucleotides **1a**–**1e** had **U_B_** units in place of thymidine residues at positions 1 (for **1a_1_**–**1a_3_**), 3 (for **1b_1_**), 7 (for **1c_1_**), 11 (for **1d_1_**), or 3 and 11 (for **1e_1_**). Since **2** has only one thymidine residue (at position 2), in addition to the corresponding **U_B_**-containing oligomer **2a_1_**, oligomers **2b_1_** and **2c_1_** were prepared in which the 2′-deoxycytidine at position 11 was replaced by a **U_B1_** unit.

### 2.2. Testing RNase H Activity against B-ASO/RNA Duplexes

Antisense oligonucleotides exert their gene silencing activity either by sequence-specific hybridization to target mRNA molecules and recruitment of RNase H, which cleaves the target mRNA (Figure 1a) or by hybridization arrest as in the splice-switching approach [23]. Here, we tested RNase H activity against B-ASO/RNA duplexes. For this, two 13-nt reference ASOs **1** and **2** were designed to be *iso*-sequential with two fragments of a previously studied 22-nt antisense oligonucleotide (ASO-22), targeting EGFR mRNA [38,42,43]. Unmodified ASOs **1** or **2** (a pyrimidine- or a purine-rich oligomer, respectively), (Table 1) were embedded in a duplex with a complementary 5′-[^32^P]-tagged RNA **3** or **4** (a purine- or a pyrimidine-rich oligomer, respectively), (Table 1), and duplexes **1/3** and **2/4** were assayed for elicitation of RNase H activity.

The content of radioactive 5′-[^32^P]-RNA in the reaction mixtures of ASO/RNA duplexes treated with RNase H was determined after 0, 5, 15, 30, 60, and 90 min by PAGE analysis (20% polyacrylamide/7 M urea). Densitometric analysis of intact 5′-[^32^P]-RNA substrate from three independent PAGE analysis experiments showed that purine-enriched ASO oligomer **2** in an ASO/RNA duplex **2/4** was readily able to activate RNase H, resulting in hydrolysis of **4** with a t_1/2_ = 11 min (Figure 1c). In contrast, pyrimidine-enriched ASO **1** had a lower potential to activate RNase H to cleave RNA **3** in the corresponding duplex **1/3**, as the half-life of this cleavage was over 90 min (Figure 1c). Interestingly, this result is supported by previously reported data that RNase H acts on native DNA substrates, preferring purine-enriched sequences compared with pyrimidine-enriched DNAs [52].

### 2.3. RNase H-Assisted Hydrolysis of RNA 3 in Duplex with B-ASOs 1 Decorated with FESAN

In the following studies, we tested the ability of RNase H-assisted hydrolysis of ^32^P-RNA **3** in its complex with the complementary B-ASO oligomers **1a_1_**, **1b_1_**, **1c_1_**, **1d_1_** and **1e_1_** (listed in Table 1), under the same conditions as for the unmodified **1**. As shown in Figure 2a, compound RNA **3** embedded in the duplexes with **1a_1_**, **1c_1_**, and **1d_1_** was almost completely degraded within the first 15 min (Figure 2a). These results suggest that the presence of a single **U_B1_** unit at the 5′- or 3′-ends or in the middle of the pyrimidine-rich B-ASO **1** strand significantly enhances enzymatic hydrolysis of the complementary RNA compared with its duplex with native ASO **1**. Additionally, incorporation of a single **U_B1_** unit at the third position of the 5′-end of the strand (as in **1b_1_**) increased RNase H activity, but to a lesser extent than in **1a_1_**, **1c_1_**, and **1d_1_**. Here, the RNA substrate **3** was completely consumed within the first 60 min. In contrast, two **U_B1_** units at positions 3 and 7, as in **1e_1_**, completely blocked RNase H activity toward complementary RNA. This result is not surprising and is likely due to steric hindrance caused by the presence of two boron cluster units in the B-ASO/RNA duplex. Interestingly, the specificity of the enzyme differed depending on the position of the boron cluster unit in the ASO strand. The **U_B1_** unit introduced into **1** at position 1, 7, or 11 produced a single cleavage product of **3** after 90 min of incubation, whereas the **U_B1_** unit at position 3 resulted in the formation of two ^32^P-RNA products (red arrows, Figure 2a).

Kinetic analysis of remaining intact ^32^P-RNA **3** showed that RNAs in duplexes with **1a_1_** and **1c_1_** were the best substrates for RNase H (t_1/2_ = 3 min, Figure 2b) and were hydrolyzed by RNase H more than 30 times faster, and in duplex with **1d_1_** (t_1/2_ = 9 min, Figure 2b) more than 10 times faster than the unmodified oligomer **1** (t_1/2_ > 90 min, Figure 2b).

### 2.4. RNase H-Assisted Hydrolysis of RNA 4 in a Duplex with B-ASOs 2 Decorated with FESAN

Next, we investigated whether the decoration of purine-rich ASO **2** with boron clusters plays a role in the kinetics of RNase H-assisted hydrolysis of complementary RNA **4**. To this end, the unit **U_B1_** containing metallacarborane was introduced at positions 2, 11, or 2 and 11 of B-ASO **2** (as in **2a_1_**, **2b_1_**, and **2c_1_**, respectively), (Table 1), and as such decorated **2** were hybridized with ^32^P-labelled RNA **4** and subjected to RNase H-assisted RNA degradation. As shown in Figure 3a, the efficiency of degradation of **4** was similar for all four tested duplexes, although the ratio of ^32^P-RNA cleavage products was slightly different depending on the labelling position. However, the general conclusion is that the presence of boron cluster decorations in the purine-rich B-ASO strand **2** has a minimal effect on RNase H activity (Figure 3b).

### 2.5. Dependence of RNase H Activity on the Nature of the Boron Cluster

To evaluate the activity of RNase H against duplex **1**/**3** decorated with different boron clusters, we incorporated the **U_B1_**, **U_B2_** or **U_B3_** units into the first position at the 5′-end of the pyrimidine-rich oligonucleotide **1**. The B-ASO/**3** duplexes containing **1a_1_**, **1a_2_**, and **1a_3_** B-ASO, respectively (Table 1), were tested for their ability to serve as RNase H substrates. In these experiments, the ratio of RNase H to ASO/RNA duplex was reduced threefold compared with the previous experiments (Materials and Methods, Section 4.2.2), while the reaction time was extended to 240 min (Figure 4a). It was previously shown that the presence of FESAN (**U_B1_**) at the 5′-end of B-ASO significantly increased the rate of RNase H-assisted hydrolysis of **3** compared with the native **1**/**3**. In the present experiments, duplexes with the **U_B1_** (FESAN) and **U_B3_** (DDB) units were better recognized and hydrolyzed than those with the **U_B2_** unit (Figure 4a,b). We discovered that the half-life of **3** in duplexes with B-ASOs corresponded with the charge of the boron cluster unit. The B-ASOs with negative charge were better models for enzyme recruitment and its nucleolytic activity compared to the non-charged B-ASO **1a_2_**. For example, RNA **3** in the duplex with **1a_3_** (DDB, charge -2) existed less than half the time (t_1/2_ = 22 min) compared to the duplex with **1a_1_** (t_1/2_ = 50 min) (FESAN, charge -1), whereas in the duplex with **1a_2_** decorated with a neutrally charged 1,2-DCDDB boron cluster (charge 0), the half-life exceeded 240 min and **3** was only slightly degraded under the same conditions. Moreover, for reaction times longer than 120 min, we observed that hydrolysis of RNA **3** in duplex with **1a_3_** (DDB, charge -2) reached an equilibrium state, in contrast to B-ASO **1a_1_** (FESAN, charge -1), in which RNA **3** substrate was completely hydrolyzed. In this experiment, we have shown that the charge and nature of the boron cluster are critical for RNase H activity. It is an extremely valuable observation for future B-ASO drug development.

## 3. Discussion

Interest in boron chemistry has steadily increased, and the use of boron clusters in medicine, especially in antiviral, antibacterial [53], and anticancer [28,36] therapies has experienced a resurgence in recent years. Due to their chemical structure, boron clusters resemble 3D capsules rich in boron atoms (^10^B), which are necessary for boron neutron capture therapy (BNCT) [48,54].

Recently, boron clusters and their conjugates with nucleosides were shown to have high binding affinity for plasma proteins under in vitro and in vivo conditions [39]. This is crucial for the pharmacodynamics of the drugs and pharmacokinetic parameters such as absorption, distribution, metabolism, and excretion (ADME) [55]. Boron clusters have been shown to interact with cationic peptides and other charged and neutral biomolecules, such as acetylcholine and amino acids, as well as vitamins, antibiotics, neuromuscular blockers, and proteins [34]. Interestingly, boron clusters can enter cancer cells as drug carriers in two ways. One of them depends on the effective transport across the membrane in vitro and in vivo [30,34,35,36]. For instance, cellular uptake of metallacarboranes were internalized 1000-fold better in HeLa cells than *closo*-dodecaborates [37]. The second pathway involves the affinity of boron clusters for receptors such as retinoid receptors [31], androgen receptor [32], vitamin D receptor [40], and the ghrelin receptor [41], being active as human immunodeficiency virus protease receptor ligands, 5-lipoxygenase receptor ligands, cyclooxygenase ligands, ligands of delocalized lipophilic cations, hyaluronic acid ligands, and histone deacetylase ligands [28].

In our previous work, we noticed that DNA nanostructures with incorporated boron clusters (1,2-DCDDB) had some ability to penetrate the phospholipid membrane of A431 cells [45]. Then, we suggested that the size of the nanostructures (<100 nm) is the feature responsible for their free cellular uptake. Recently, we have shown that an antisense oligonucleotide decorated with boron clusters (FESANs) is effectively taken up by EGFR-overexpressing cells via the EGF receptor (data not shown). The pools of proteins associated with two tested boron clusters (1,2-DCDDB or FESAN) are very similar and contain a very specific group of proteins from whole cell protein lysates. Therefore, we suggest that boron clusters own the affinity for EGFR, demonstrating as important feature for the transport of drugs into the cancer cells (will be published soon). Recently, a conjugate of COSAN and a Tβ4 protein (thymosin β4) incubated with cardiomyocytes was shown to increase the viability of rat hypoxic cardiac cells and enhanced the wound healing potential of Tβ4 in human fibroblasts [33]. Our studies have also shown that the presence of metallacarborane (FESAN) associated with either ASO (B-ASO) or a nucleoside moiety increases the mitochondrial activity of cells by up to ~20% or ~40% (MTT assay) [42,43].

ASOs have been studied as cancer therapeutics for decades with promising in vitro results, and many of them have been tested in clinical trials [56]. However, there are currently no approved oligonucleotide drugs in oncology. Three ASOs have received an orphan drug designation (Oblimersen for chronic lymphocytic leukemia, PNT2258 for diffuse large B-cell lymphoma, and Cobomarsen for mycosis fungoides cutaneous T cell lymphoma). Most oligonucleotide anticancer drugs are still in first or second phase of clinical trials and are looking for new “smart” strategies to be applicable [15].

Previously, we showed that 13-nt ASO models modified with negatively charged FESAN residues (−1) provided interesting affinity data for crude snake venom (svPDE) exonuclease [38]. Inspired by these results, we decided to use two short (13-nt) natural (unmodified) DNA oligonucleotides **1** and **2**, which differed by their nucleoside composition as antisense oligonucleotides (ASOs). Oligonucleotide **1** was a pyrimidine-rich oligomer and oligonucleotide **2** was a purine-rich oligomer. These two ASO models were modified with different boron clusters, including (−2) negatively charged DDB, (−1) negatively charged FESAN, and (0) neutrally charged 1,2-DCDDB, resulting in a series of B-ASOs **1a_1_**, **1a_2_**, **1a_3_**, **1b_1_**, **1c_1_**, **1d_1_**, **1e_1_** and **2a_1_**, **2b_1_**, **2c_1_** listed in Table 1. The aim of this study was to test the hypothesis of whether oligonucleotides modified with boron clusters with different charges and in different positions of the ASO chain affect the nucleolytic activity of RNase H, which hydrolyses RNA substrates in B-ASO/RNA duplexes, leaving RNA products with a 3′-hydroxyl group and a 5′-phosphate group. Explanation and confirmation of this phenomenon would be critical for further research and development of B-ASOs for therapeutic use.

First, we showed that RNase H activity toward ASO/RNA duplex was higher in the case of **2/4** (t_1/2_ = 11 min, Figure 1c) than duplex **1/3** (t_1/2_ time > 90min, Figure 1c), suggesting that purine-enriched ASO oligomer **2** was a better component for the antisense approach than pyrimidine-enriched ASO **1**. The RNA in the **1/3** duplex was stable in a 90 min incubation, whereas the RNA in the **2/4** duplex was completely degraded after 30 min and gave three different RNA products in the 15, 30, and 90 min of the reaction courses (Figure 1b). A similar RNase H activity depending on the nucleotide composition of ASO was reported by Ho et al., who reported that A- and T-rich ASOs can hybridize to their RNA target sites, resulting in an ASO/RNA duplex with lower thermodynamic stability, which decreases efficient processing by RNase H [52]. Kiełpiński and co-authors confirmed that the sequence preferences of *E. coli* RNase H and human enzymes are almost identical [57]. Both endonucleases prefer G- or C-rich RNA sequences, with C/G, C/G/U or A/U sequences upstream of the cleavage site. In addition, at least four ribonucleotides upstream and two ribonucleotides downstream of the cleavage site are required for optimal cleavage of these RNase H motifs [57]. Tu et al. observed that only ASOs molecules with a tetranucleotide motif TCCC (complementary to GGGA on RNA) yielded effective antisense oligonucleotides against tumor necrosis factor (TNF-alpha) [58]. An antisense strand (13 of 22 nt) containing more than 50% of the TCCC motif effectively inhibited TNF-alpha synthesis [58].

In our activity experiments to demonstrate the dependence of RNase H on the FESAN position in ASO oligomer **1**, we designed five B-ASOs (**1a_1_**, **1b_1_**, **1c_1_**, **1d_1_**, and **1e_1_**) and confirmed that the presence of a single FESAN modification at the 5′-, 3′-end or in the middle position of pyrimidine-rich **1** increased the ratio of the enzymatic hydrolysis of complementary RNA **3** by up to 30-fold compared to unmodified **1**. In all cases, a radioactive ^32^P-RNA hydrolysis product was eventually generated (Figure 2). These surprising results suggest that the presence of metallacarborane in the pyrimidine-enriched ASO oligomer backbone alters the recruitment of B-ASO/RNA and ultimately the antisense activity of ribonuclease RNase H. It is worth noting that the location of the boron cluster closer to the 5′-end of B-ASO (as in **1a_1_** and **1b_1_**) allows RNase H to hydrolyze RNA at two cleavage sites, whereas the location of **U_B1_** in the middle or 3′-end of the ASO strand restricts RNA cleavage to one position.

RNase H cuts the RNA of the hybrid duplex near the 5′-end of ASO [56,59], so incorporation of a modification at this position may play a key role in the recognition and binding of the B-ASO/RNA duplex by RNase H. As reported by Crooke et al., all proteins preferentially bind to the 5′-end of ASO, which is the major docking site, although this effect may be different for specific ASO sequences and for individual proteins [60]. The sitting of two FESANs units at positions 3 and 7 of the ASO strand, as in **1e_1_**, interferes with the recognition of the target duplex by RNase H. This effect can be observed in the ASO strand and attributed to the steric hindrance of the boron cluster-modified **1e_1_/3** duplex, which alters the interactions with the binding pocket of RNase H.

Contrary to our assumptions, the introduction of the FESAN moiety at positions 2 or 11, and simultaneously at 2 and 11 positions into purine-rich oligomer **2** as in **2a_1_**, **2b_1_**, and **2c_1_** had little or no effect on the activity of RNase H (Figure 3a,b) toward RNA substrate **4**. The observed comparable nucleolytic activity of RNase H for duplexes of all **2** oligomers with **4** for all these models of **2/4** suggests that the enzyme binding core binds preferentially to purine-rich ASO, regardless of decoration by the boron cluster. In addition, two recent results (Figure 2a and Figure 3a) support the previously obtained data [43] that 22-mer ASO (from upstream sequences covering **1** (underlined) and downstream sequences covering **2** (italic), of sequence 5′-d(TTT CTT TTC *C**T**C CAG AGC CCGA*)-3′), heavily decorated with FESAN in the 5′-domain (4 or 5 FESANs moieties within sequence **1**), caused the same degradation of EGFR mRNA as native ASO (without FESAN decoration). When testing the library of B-ASO compounds (of the same 22-mer sequence), we selected the most active ASOs containing a single boron cluster in position 11 (highlighted in bold), corresponding to the B-ASO-**1d_1_**/RNA duplex [42].

Our results support the half-life data presented in Figure 4, where we showed that the hydrolytic activity of RNase H depends on the type of boron cluster and its charge. As the negative charge of the boron cluster increases, the rate of RNA hydrolysis by RNase H increases in the following order of boron cluster charge (0) 1,2-DCDDB < (−1) FESAN < (−2) DDB, (t_1/2_ = 240 min < 50 min < 22 min, respectively). Interestingly, our previous work dealing with the silencing activity of B-ASOs achieved by the antisense approach towards the EGFR mRNA reached a maximum of 83–95% with the double negative boron cluster [(−2) DDB] in HeLa cervical cancer cells [42]. The work of Kwiatkowska et al. confirmed that the silencing activity of B-siRNA containing a neutral charge of the boron cluster (1,2-DCDDB) was basically unchanged compared with the unmodified reference siRNA duplex [46]. The only difference observed in thermodynamic stability and silencing activity was attributed to siRNA duplexes possessing a boron cluster (1,2-DCDDB) at the 5′-end of the antisense strand, although in this case the silencing mechanism was based on RNA interference rather than antisense RNase H action. Moreover, extending the reaction time to 240 min (Figure 4a,b) showed complete hydrolysis of RNA substrate **3** in the duplex **1a_1_** (FESAN, charge -1), in contrast to duplex **1a_3_/3** (DDB, charge -2), where a low amount of intact RNA **3** remained even after 240 min of the hydrolysis reaction. Initial reaction times appear to be critical for RNase H activity and effectiveness in the presence of B-ASO, as confirmed by our B-ASO/RNA hydrolysis analyses (Figure 4) and previous biological studies using the dual fluorescence assay (DFA) [42]. As shown by Goszczyński and co-workers, metallacarboranes (COSAN and FESAN) exhibit the strongest interaction with albumin among the tested clusters in the order of: metallacarboranes [M(C_2_B_9_H_11_)_2_]^−^ > carboranes (C_2_B_10_H_12_) >> dodecaborate anion [B_12_H_12_]^2−^. Accordingly, metallacarboranes first interact specifically with the binding cavity of albumin and then, with increasing concentration, interact non-specifically with the protein surface [39]. Kodr and co-authors showed that the modified 2′-deoxyribonucleoside triphosphates (dNTPs) with boron clusters (FESAN, COSAN or dicarba-*nido*-undecaborate) were efficient substrates for DNA polymerases (KOD XL DNA) despite very large and lipophilic boron substituents and were used in the enzymatic synthesis of modified B-DNA [61].

Recently, we confirmed that the ASO strands in the B-DNA nanostructure, despite their binding to the boron cluster (1,2-DCDDB), are able to hybridize with the target RNA sequence and lead to RNase H-assisted RNA hydrolysis at a slightly higher rate than that of the unmodified ASO-22/RNA [45]. Leśnikowski and co-workers showed that incorporation of uncharged dodeca(thymidine phosphates) containing 5-(o-carboranyl-1-yl)-2′-deoxyuridine units (CDU) or the carboranyl group in nido-1 form with a negative charge into the 12-nt oligomer was more efficiently digested into the central position (6th position) by RNase H, however, the results of this study have not been explained sufficiently [62].

As described by Østergaard et al., RNase H makes several specific contacts with the phosphate scaffold on the DNA strand of the heteroduplex. Of particular note is an electrostatic interaction of RNase H with the DNA strand in the phosphate-binding pocket [63]. The structure of the hybrid binding domain (HBD) is unique relative to affinity with the DNA/RNA substrate of RNase H. Cerritelli et al. have shown that the RNA binding loop consisting of five amino acid residues (D51, R52, F53, P54, A55) is supported by the highly conserved R35, which forms a hydrogen bond with the carbonyl oxygen of D51 [24,64]. The main chain amides of R52 and A55 form hydrogen bonds with the 2′-OH groups of RNA (mRNA substrate). The presence of the groove contains aromatic rings of W43 and F58 interacting with three phosphodeoxyribose units of the DNA strand. The hydroxyl of a highly conserved Y29, the amine and amide of K60, and the guanidinium of R57 interact with three consecutive phosphates of DNA (ASO strand) of the heteroduplex. We hypothesize that boron clusters behave like a “magnet” and electrostatically attract the RNase H-binding pocket (positively charged amino acid K60) to B-ASO (negatively charged). Thus, when the negative charge of the boron cluster increases the RNase H activity is enhanced. On the other hand, the hydrophobic nature of the boron cluster and the presence of hydrophobic amino acids (W43, F53, P54, A55, F58) in HBD favor the binding efficiency and the interaction of these two molecules.

In our work, metallacarborane (FESAN) was clearly shown to have appropriate affinity for this single-stranded DNA and this was independent of its sequence as measured by the MST [38]. By decorating ASO with metallacarboranes (FESANs), we can likely achieve an increased affinity of B-ASO for target mRNA in cancer cells that dynamically change their spatial structure from single-stranded to hairpin. Given the relatively low copy number of most mRNAs in cells [65], we hypothesize that consideration of the nature and position of boron clusters in the design of ASO sequences will result in more efficient cleavage of mRNA by RNase H. Optimization of B-ASO structures with chemical modifications (related to increased nucleolytic stability and hybridization affinity) will deepen the development of the “smart” B-ASO strategy [research in progress].

## 4. Materials and Methods

### 4.1. Chemistry

Standard nucleoside phosphoramidites and 5′-dimethoxytrityl-2′-propargyluridine 3′-*O*-(*N,N*-diisopropyl-2-cyanoethyl) phosphoramidite were purchased from ChemGenes Corporation (Wilmington, MA, USA). [(3,3′-Iron-1,2,1′,2′-dicarbollide)^−1^]ate (-1) cesium salt (ferra(III) bis(dicarbollide), ([C_2_B_9_H_11_)^2^]^−^), FESAN), was purchased from Katchem (Režn/Prague, Czech Republic). Radiolabeled adenosine 5′-triphosphate ([γ-^32^P]-ATP) was purchased from Hartmann-Analytik (Braunschweig, Germany), and T4 polynucleotide kinase and kinase buffer were purchased from BioLabs (Ipswich, MA, USA). Autoradiography double-coated films Medical X-ray blue (MXBE film) were purchased from Carestream (New York, NY, USA). The developer reagent and fixer reagent were purchased from Kodak Processing Chemicals (Sigma Aldrich, St. Louis, MO, USA). RNase H (Ribonuclease H, *E. coli*), an RNA-specific endonuclease that cleaves the RNA within RNA/DNA hybrids was purchased from EURx, (Gdańsk, Poland).

#### 4.1.1. Automated Synthesis of Oligonucleotides

Alkyne-functionalized DNA oligonucleotides were synthesized according to the phosphoramidite solid-phase approach using a solid support LCA CPG and commercial phosphoramidites of T, dC, dA, dG, and 2′-*O*-propargyl-5′-DMT uridine with DMT protecting groups on their 5′-hydroxyl moieties, and their exocyclic amine functions were protected with acetyl, benzoyl, or iso-butyryl groups for dC, dA, and dG, respectively. The commercially available nucleoside-linked solid support LCA-CPG and 0.07 M solutions of the monomers in CH_3_CN were used. Oligonucleotides were synthesized at 0.1 µmol scale using an automated DNA/RNA synthesizer H6 GeneWorld (K&A, Laborgeraete GbR, Schaafheim, Germany) under standard conditions for DNA oligonucleotide synthesis [66]. All compounds were cleaved from the solid support as 5′-DMT-protected derivatives and purified by reversed-phase high-performance liquid chromatography (RP-HPLC) (“DMT-on” mode). Deprotection of the 5′-OH groups was performed on C18 SepPak cartridges (Waters, Milford, Ireland) containing 2% trifluoroacetic acid (TFA). The sequence and purity of the compounds obtained were confirmed by matrix-assisted laser desorption/ionization coupled to time-of-flight mass spectrometry (MALDI-TOF MS) or electrospray-ionization quadrupole time-of-flight (ESI-Q-TOF) mass spectrometry, analytical RP-HPLC and denaturing polyacrylamide gel electrophoresis (20% PAGE, 7 M urea). The analyzes were described in the previous work [38]. Unmodified oligonucleotides **1** and **2** and their complementary oligonucleotides **3** and **4** were obtained by an analogous method using a routine DNA/RNA oligonucleotide synthesis procedure. The RP-HPLC chromatograms and mass spectra of oligomers **1**, **2**, **3**, **4**, **P1a**-**P1e** and **P2a**-**P2c**, are shown in Appendix A.

#### 4.1.2. Synthesis of Oligonucleotides 1a–1e and 2a–2c Modified with Boron Clusters

Oligonucleotides **P1a**, **P1b**, **P1c**, **P2a** and **P2b** were post-synthetically modified with one of the following azide derivatives of boron clusters: ([(3,3′-Iron-1,2,1′,2′-dicarbollide)^−1^], N_3_-alkyl-FESAN, [Fe(C_2_B_9_H_11_)_2_]^−^) (**B_1_**), 1,2-dicarba-*closo*-dodecaborane (N_3_-alkyl-1,2-DCDDB, [C_2_B_10_H_12_]) (**B_2_**) and dodecaborate (N_3_-alkyl-DDB, [B_12_H_12_]^2−^) (**B_3_**) (Appendix A) equipped with an alkyl azide (N_3_-alkyl) function suitable for the alkyne–azide click reaction. Reactions were performed according to the standard copper sulfate procedure used in the chemistry of oligonucleotides [67]. First, the following solutions were prepared: (1) DMSO (Sigma-Aldrich, St. Louis, MO, USA)/tBuOH (Sigma-Aldrich, St. Louis, MO, USA) (3:1, *v*/*v*), (2) 0.1 M copper sulfate in water, (3) 0.2 M tris(3-hydroxypropyltriazolylmethyl) amine (THPTA) in DMSO/tBuOH and (4) 520 mM solutions of azides 6–8 (Figure 1) in DMSO (Sigma-Aldrich, St. Louis, MO, USA)/tBuOH (Sigma-Aldrich, St. Louis, MO, USA) (3:1, *v*/*v*). Then, the THPTA and copper sulfate solutions were mixed in a 1:1 ratio and degassed in an argon stream. Fifty molar equivalents of the alkyl azide-coupled boron cluster (3.34 µL) and 25 molar equivalents of the copper sulfate/THPTA solution (9.19 µL) were added to a vial containing the oligonucleotide (4 OD) in water (40 µL). The mixture was vortexed, and when a precipitate formed, approximately 80 µL of DMSO (Sigma-Aldrich, St. Louis, MO, USA)/tBuOH (Sigma-Aldrich, St. Louis, MO, USA) (3:1, *v*/*v*) was added. Then, 40 equivalents of freshly prepared 0.1 M sodium ascorbate in water (14.4 µL) was added to the reaction mixture. The final solution was briefly degassed with a stream of argon and stirred for 4 h at room temperature [38,42,43].

Subsequently, the products were isolated by RP-HPLC using a Kinetex 5 µm, C_18_, 250 mm × 4.6 mm column with buffer A (0.1 M CH_3_COONH_4_/H_2_O) and buffer B (100% CH_3_CN) at a flow rate of 1 mL/min. The buffer gradient was as follows: (1) 0–2 min 0% buffer B; (2) 2–25 min 0–48% buffer B; (3) 25–30 min 48–60% buffer B; (4) 30–35 min 60–0% buffer B; and (5) 35–38 min 0% buffer B (details can be found in Table 1 or in Appendix A). UV detection was performed at λ_max_ = 268 nm. The collected oligonucleotide fractions were desalted on C18 SepPak cartridges (Waters). The molecular mass of each synthesized oligonucleotide was confirmed by MALDI-TOF MS or ESI-Q-TOF mass spectrometry (Table 1, Appendix A).

### 4.2. Enzymatic Assays

#### 4.2.1. 5′-Phosphorylation of RNA Oligonucleotides

A 0.10 OD RNA **3** (0.59 nmol) or **4** (0.72 nmol) (solution in 15 µL of Milli-Q water was mixed with ^32^P-radiolabeled ATP (37.0 MBq, 1.00 mCi, diluted 10x with Milli-Q water, 2 µL), T4 polynucleotide kinase (Lab-JOT, Warsaw, Poland) (1 µL, 10 unit/µL,100mM MgCl_2_, 50 mM DTT) and 2 µL of manufacturer’s kinase 10x reaction buffer (700mM Tris-HCl (pH 7.6 at 25 °C). The reaction mixture, with a total volume of 20 µL, was incubated for 1 h at 37 °C. The mixture was then inactivated by incubation at 80 °C for 3 min and used for further studies without additional purification.

#### 4.2.2. In Vitro RNase H-Assisted RNA Cleavage

Synthesis of RNA oligonucleotides at the 0.1 µmole scale was performed according to routine procedures on a Gene World DNA synthesizer H6 GeneWorld (K&A, Laborgeraete GbR, Schaafheim, Germany) under the conditions recommended by the manufacturer. All compounds were cleaved from the solid support, deprotected using standard procedures, and then purified by RP-HPLC [66]. In the experiment, ^32^P-radiolabeled substrate RNA **3** or **4** (2 µL of the stock solution after phosphorylation) was mixed with **1a–1e** (0.05 OD, 0.45 nmol, 5 µL) or **2a–2c** B-ASO (0.05 OD, 0.37 nmol, 5 µL) in 20 mM Tris-HCl buffer (pH 8) containing 50 mM NaCl and 10 mM MgCl_2_ in a total volume of 7 µL. The mixture was heated at 75 °C for 5 min and slowly cooled (1 h) to room temperature. Reaction buffer containing 200 mM Tris-HCl, 500 mM KCl, 200 mM DTT, 50 mM MgCl_2_ (3 µL) and Milli-Q water (19 µL) was added 10x to each sample at 4 °C, and then ribonuclease H (RNase H, Eurx, Poland) (1 µL of 0.3 U/µL stock solution for Figure 1, Figure 2 and Figure 3, or 1 µL of 0.1 U/µL stock solution for Figure 4) was added to a total volume of 30 µL. The resulting assay mixture was incubated at 37 °C for up to 90 min. Aliquots of the enzymatic reaction (3 µL) were removed from the reaction mixture at predetermined time points (0, 5, 15, 30, 60, 120, and 240 min) and mixed with loading buffer (10 mM Tris-HCl, 60 mM EDTA, 60% glycerol, 0.03% bromophenol blue, 0.03% xylene cyanol, pH 7.6) (6 µL), and the enzyme was inactivated by incubating the reaction sample at 80 °C for 3 min. Each sample was analyzed by 20% PAGE with 7 M urea at room temperature and 20 mA for 1 h. The enzyme was inactivated. After completion of electrophoresis, the gels were transferred to an exposure cassette and covered with double-coated autoradiography films (MXBE film, Rochester, NY, USA) for 10 min at low temperature (−25 °C). Then, the double-coated films were soaked in the developing reagent (Kodak Processing Chemicals, Sigma Aldrich, St. Louis, MO, USA) and then in the fixing reagent and scanned using a G-box instrument (Syngene, Cambridge, UK). Autoradiographic analyzes of intact 5′-[^32^P]-RNA substrates were performed using three independent PAGE analysis experiments (Figure 1, Figure 2, Figure 3 and Figure 4).

## 5. Conclusions

Therapeutic nucleic acids hold great promise as gene silencing agents due to their functionality and diverse applicability. Recently, during the global pandemic caused by the SARS-CoV-2 virus, new strategies based on ASOs have been developed, providing hope for effective disease control.

In this work, we have deepened the scientific understanding of the properties of boron clusters and demonstrated that they can enhance the efficiency of recognition and hydrolysis of B-ASO/RNA heteroduplexes recruited by the endonuclease RNase H. This phenomenon was demonstrated on single-stranded DNA oligonucleotides enriched in pyrimidine and coupled to FESAN. The rate of hydrolysis of the RNA substrate depended on the nucleotide sequence and the position of the modification with the boron cluster. The affinity of FESAN for proteins, as already shown in the literature [38], confirmed by us on the crude mixture of venom proteins, on ssDNA, and confirmed here on short B-ASO oligonucleotides, explains its involvement in the attraction of the two reacting biomolecules and accelerates the reaction rate between heteroduplex B-ASO/RNA and enzyme protein (RNase H). Moreover, we confirmed that the nature of the boron cluster and its charge are crucial for enhancing the activity of RNase H and the rate of hydrolysis of the RNA substrate (being a therapeutic target). This discovery is of enormous importance for the use of boron clusters in biological processes where oligonucleotides and cellular proteins are required for biological effects, such as enhancing of the therapeutic effect of nucleic acids. In our laboratory, the affinity of boron clusters for enzymatic and cellular proteins is being studied. These results could be important for the development of therapeutic boron cluster nucleic acids for various diseases, as they offer to reduce the therapeutic dose without losing their high efficacy.

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
