# Peer review of "Boron Clusters as Enhancers of RNase H Activity in the Smart Strategy of Gene Silencing by Antisense Oligonucleotides"

_ijms, 2022, doi:10.3390/ijms232012190_

Round 1

Reviewer 1 Report

The manuscript "Boron clusters as enhancers of RNase H activity in the smart strategy of gene silencing by antisense oligonucleotides" submitted to International Journal of Molecular Sciences by Kaniowski D. et al. devoted to the investigation of the influence of boron clusters attached to the antisense oligonucleotides on the RNase H activity.

The aim of this article is clear from the abstract. It is not clear from the text what does "smart" strategy of gene silencing mean. Is it some new approach? The title is not relevant.

In the introduction authors give us the information about antisense strategy of gene inhibition and RNase H dependent way of the mRNA destroying.  Methods are described in details in Materials and Methods. The authors have used appropriate methods. The tables and figure is relevant and clearly presented. The results are discussed and the conclusions are logic and clear. The study design corresponds to the aim of work.

I need to indicate same minor points of this manuscript.

1. The new concept ("smart" strategy) proposed by author must be presented more detailed in manuscript.

2. Figure 2. For duplexes ASO 1c1/RNA 3 the time ½ is 3 min, but if we compare the data 1ca with 1b1 and 1d1 it must be much more, isn't it?  

3. In the section Materials and Methods (4.2.2) the description of RNA synthesis is absent. Please add this information in manuscript.

4. In the section Materials and Methods the concentrations of RNA and ASO are absent. Please add the concentration and ratio of components in reaction mixture.

I would suggest that this manuscript can be published after Minor Revisions.

Author Response

Response to Reviewer #1

Reviewer: The manuscript "Boron clusters as enhancers of RNase H activity in the smart strategy of gene silencing by antisense oligonucleotides" submitted to International Journal of Molecular Sciences by Kaniowski D. et al. devoted to the investigation of the influence of boron clusters attached to the antisense oligonucleotides on the RNase H activity.

The aim of this article is clear from the abstract. It is not clear from the text what does "smart" strategy of gene silencing mean. Is it some new approach? The title is not relevant.

In the introduction authors give us the information about antisense strategy of gene inhibition and RNase H dependent way of the mRNA destroying.  Methods are described in details in Materials and Methods. The authors have used appropriate methods. The tables and figure is relevant and clearly presented. The results are discussed and the conclusions are logic and clear. The study design corresponds to the aim of work.

I need to indicate same minor points of this manuscript.

I would suggest that this manuscript can be published after Minor Revisions.

Response: We thank the reviewer for all the positive comments and detailed review of our manuscript. We have taken into account all your comments and valuable suggestions, which have helped us to improve our manuscript considerably. Below we present our corrections to each of the reviewer's comments. Fragments in red are introduced to the manuscript text.

  1. The new concept ("smart" strategy) proposed by author must be presented more detailed in manuscript.

Response: We have added the following paragraph in the Introduction:

Lines 99-102: New "smart" chemical strategies are constantly being sought to enhance the activity and efficacy of therapeutic nucleic acids.  Recently, boron clusters have received special attention because of their beneficial biomedical functions as pharmacophores [28,29], "enhancers" of drug action [30-33], membrane carriers [34-37], protein binders [38,39] and receptor ligands [31,32,40,41].

The respective references were aligned accordingly, and one more reference was added:

[29] Ali F, S Hosmane N, Zhu Y. Boron Chemistry for Medical Applications. Molecules. 2020 Feb 13;25(4):828. doi: 10.3390/molecules25040828. PMID: 32070043; PMCID: PMC7071021.

  1. Figure 2. For duplexes ASO 1c1/RNA 3 the time ½ is 3 min, but if we compare the data 1ca with 1b1 and 1d1 it must be much more, isn't it?

Response: In Fig. 2, 1c1/3, the product on the second lane corresponds to the first RNA degradation product, which is slightly shorter than the initial RNA. But the starting RNA is already degraded (T1/2 = 3 min). After another 10 min (lane 3), the next degradation product is then visible. These two products are now indicated by the red arrows. This means that the half-life has been correctly estimated. T1/2 for duplex 1c1/3 is shorter that for 1d1/3. 

We would like to add that we repeated this experiment three times and the result confirmed this fast kinetics of the RNA cleavage, when complexed with 1c1. This is not a technical error.

  1. In the section Materials and Methods (4.2.2) the description of RNA synthesis is absent. Please add this information in manuscript.

Response: We have added description of RNA synthesis (4.2.2) as suggested.

“Synthesis of RNA oligonucleotides at the 0.1 µmole scale was performed according to routine procedures on a Gene World DNA synthesizer H6 GeneWorld (K&A, Laborgeraete GbR, Schaafheim, Germany) under the conditions recommended by the manufacturer. All compounds were cleaved from the solid support, deprotected using standard procedures, and then purified by RP-HPLC [66].”

  1. In the section Materials and Methods the concentrations of RNA and ASO are absent. Please add the concentration and ratio of components in reaction mixture.

Response: Corrected as suggested.

Reviewer 2 Report

1. Section 2.1. Methods of synthesis (first two paragraphs) should be described in brief.

2. Lines 178-180 – these sentences should be moved to the Discussion.

3. Lines 489-493. Brief description of the abovementioned methods is needed here.

4.The sources of some reagents are not indicated (tBuOH, DMSO).

5. Lines 520-522 – what means “0% B”? It is not explained. Is it buffer B?

6. Line 530 – the composition of manufacturer's kinase buffer or reference should be included.

Author Response

Answers for Reviewer #2

We thank the Reviewer for the positive comments and detailed review of our manuscript. We have taken into account all the comments and valuable suggestions, and they have helped us to improve our manuscript considerably. Below we present our corrections to each of the reviewer's comments. Fragments in red are introduced to the manuscript text.

  1. Section 2.1. Methods of synthesis (first two paragraphs) should be described in brief.

Response: In our opinion, Section 2.1, Synthesis Methods (the first two paragraphs) in the Results section is clearly written and contains sufficient general information about the design of the oligonucleotides and the synthesis approach used. The first paragraph describes the lead oligonucleotide ASO 22-mer anti-EGFR (DNA), from which two shorter 13-nt DNA fragments were generated (1 and 2, homopyrimidine and a purine-rich oligomer, respectively). Subsequently, routinely synthesized 2'-O-propargyluridine (UPr) units were decorated with boron clusters (one of the three types of boron clusters) in a post-synthetic click reaction (which just was awarded by a Nobel Prize!). The detailed method for synthesis of all boron-labeled oligonucleotides is described in the Materials and Methods section.

  1. Lines 178-180 – these sentences should be moved to the Discussion.

Response: Done as suggested.

  1. Lines 489-493. Brief description of the abovementioned methods is needed here.

Response: We added a reference 65 on the routine DNA synthesis at line 491.

4.The sources of some reagents are not indicated (tBuOH, DMSO).

Response: Added accordingly.

  1. Lines 520-522 – what means “0% B”? It is not explained. Is it buffer B?

Response: We have corrected it accordingly. 0% B means 0% of buffer B.

  1. Line 530 – the composition of manufacturer's kinase buffer or reference should be included.

Response: We have corrected it accordingly.

Reviewer 3 Report

This manuscript by Damian et al. describes the use of boron cluster-conjugated antisense oligonucleotides to enhance the RNase H activity to degrade target mRNA. From these studies, the authors found that the charge, nature of the boron cluster, and the position of the sequence are critical for enhancing the RNase H activity. The boron cluster-conjugated antisense oligonucleotides are obtained by click reactions and characterized by ESI and MALDI spectroscopy. Overall, the current study revealed that the metallacarborane conjugation to the oligonucleotide enhances the recruitment of RNase H to the duplex and its enzymatic activity.

In my opinion, this report would be suitable for publication but not in its current form, which should be addressable in a revision:

1. N3-alkyl-FESAN should be written as N3-PEG-FESAN.

2. In Fig2, the Autoradiodiogram of PAGE for ASO1d1 is showing efficient degradation than that of ASO1c1, but why was the half-life obtained in the opposite way?

3. It seems that the dual boron cluster labeling of ASO1e1 and ASO2c1 behaved differently on ASO1 and ASO2, respectively. Can the authors explain the reason for this?

Author Response

Answers for Reviewer #3

We thank the Reviewer for positive comments and such a detailed review of our manuscript. We have taken all comments and valuable suggestions into account, and it helped to improve our manuscript considerably. Below we present our answers/corrections to each of the reviewer's comments.

  1. N3-alkyl-FESAN should be written as N3-PEG-FESAN.

Response: Polyethylene glycol (PEG) has a structure commonly expressed as H-(O-CH2-CH2)n-OH, where "n" ranges from 4 to 120. In the case of boron cluster azides, they contain an alkyl bond terminated with the amino group (-N3), where "n" is less than 4 (1 glycol residue in FESAN derivative). The name N3-alkyl-boron cluster is correct and corresponds to the chemical nomenclature.

  1. In Fig2, the Autoradiodiogram of PAGE for ASO1d1 is showing efficient degradation than that of ASO1c1, but why was the half-life obtained in the opposite way?

Response: In Fig. 2a, 1c1/3, the product on the second lane corresponds to the first RNA degradation product, which is slightly shorter than the initial RNA. After another 10 min (lane 3), the next degradation product is then visible. These two products are now indicated by the red arrows. This means that the half-life has been correctly estimated. T1/2 for duplex 1c1/3 is shorter that for 1d1/3.

We would like to add that we repeated this experiment three times and the result confirmed this fast kinetics of the RNA cleavage, when complexed with 1c1.

  1. It seems that the dual boron cluster labeling of ASO1e1 and ASO2c1 behaved differently on ASO1 and ASO2, respectively. Can the authors explain the reason for this?

Response: Oligonucleotide B-ASO 1e1 contains two FESAN units at positions 3 and 7, with a gap of 3 nucleotides between the inserted modifications. RNase H is able to recognise any DNA/RNA heteroduplex longer than 5 base pairs. The incorporated modifications at positions 3 and 7 present a steric hindrance to the endoribonuclease. In the case of 2c1, the gap between the two modifications is 8 nucleotides, allowing RNase H to be recruited.